# Immuno-Mediated Inflammation in Hypertensive Patients with 1-h Post-Load Hyperglycemia

**DOI:** 10.3390/ijms231810891

**Published:** 2022-09-17

**Authors:** Maria Perticone, Raffaele Maio, Simona Gigliotti, Franco Arturi, Elena Succurro, Angela Sciacqua, Francesco Andreozzi, Giorgio Sesti, Francesco Perticone

**Affiliations:** 1Department of Medical and Surgical Sciences, Magna Graecia University of Catanzaro, 88100 Catanzaro, Italy; 2Azienda Ospedaliero-Universitaria Mater Domini, Catanzaro, 88100 Catanzaro, Italy; 3Department of Health Sciences, Magna Graecia University of Catanzaro, 88100 Catanzaro, Italy; 4Department of Clinical and Molecular Medicine, La Sapienza University of Rome, 00189 Rome, Italy

**Keywords:** inflammation, hypertension, cardiovascular risk, glucose tolerance

## Abstract

Inflammation plays a key role in the pathogenesis/progression of atherosclerosis, and inflammatory molecules contribute to the progression of cardiovascular disease. Subjects with normal post-load glucose tolerance and 1-h post-load plasma glucose >155 mg/dL have an increased risk of subclinical target organ damage and incident diabetes. We aimed to test possible differences in immune-mediated inflammatory parameters in newly-diagnosed hypertensives with or without 1-h post-load hyperglycemia. We enrolled 25 normotensives (NGT) and 50 hypertensives normotolerant on oral glucose tolerance test, further divided into two groups based on 1-h post-load plasma glucose: NGT 1-h ≥ 155 (*n* = 25) and NGT 1-h < 155 (*n* = 25). We measured toll-like receptor (TLR) 2, TLR4, nuclear factor kβ (NF-kβ), interleukin (IL)-1β, IL-6, IL-8, IL-10, and tumor necrosis factor (TNF)-α. Hypertensives showed significantly worse metabolic and lipid profiles, and higher values of body mass ass index (BMI), creatinine, and inflammatory parameters, compared to controls. NGT 1-h ≥ 155 had a worse glycometabolic profile and higher values of TLR2 (9.4 ± 4.2 vs. 5.9 ± 2.6 MFI), TLR4 (13.1 ± 3.9 vs. 7.8 ± 2.3 MFI), NF-kβ (0.21 ± 0.07 vs. 0.14 ± 0.04), IL-1β (6.9 ± 3.4 vs. 3.2 ± 2.1 pg/mL), IL-6 (10.8 ± 2.6 vs. 4.1 ± 1.6 pg/mL), IL-8 (27.6 ± 9.3 vs. 13.3 ± 5.6 pg/mL), TNF-α (6.4 ± 2.9 vs. 3.3 ± 1.4 pg/mL), and high-sensitivity C-reactive protein (hs-CRP) (4.8 ± 1.5 vs. 2.7 ± 1.0 mg/dL) in comparison with NGT 1-h < 155. Matsuda-index and 1-h post-load glycemia were retained as major predictors of TLRs and NF-kβ. These results contribute to better characterizing cardiovascular risk in hypertensives.

## 1. Introduction

It is well established that inflammatory processes play a key role in the pathogenesis and progression of the atherosclerotic disease and its related clinical complications [1,2,3,4]. In fact, the activation of inflammatory pathways, as demonstrated by biological and clinical evidence, are implicated in early atherogenesis, in the progression of vascular lesions and, finally, in thrombotic complications. In keeping with this, some results, obtained from observational and/or interventional trials, have demonstrated that inflammatory markers may play a major adjunctive role in the assessment of cardiovascular (CV) risk in selected patients, and may be considered as emergent therapeutic targets [2,5,6].

Similarly, CV risk factors are associated with an increase in oxidative stress and an elevation of some pro-inflammatory agents indicating the response of the vascular wall to injury as a defense reaction that, mechanistically, promotes both proliferation and expansion of smooth muscle cells in the intimal space [7]. In accordance with this, increasing evidence demonstrates that this inflammatory status, firstly driven by some CV risk factors [8,9,10] is maintained by other mechanisms, such as the activation of the immune system [11,12,13].

In the last few years, we and others have contributed to defining a specific phenotype characterized by a post-load normal glucose tolerance (NGT) but with 1-h post-load plasma glucose ≥155 mg/dL (NGT 1-h ≥ 155) [14,15,16,17]. This phenotype has higher subclinical organ damage, such as cardiac hypertrophy and diastolic dysfunction [18,19], increased vascular stiffness [20], and reduced renal function [21]; in addition, it is characterized by an increased risk of developing incident diabetes mellitus [22].

Nevertheless, to our knowledge, at this time, no data exist about a different activation of immune-mediated inflammation in NGT 1-h >155 subjects in comparison with NGT 1-h < 155. Thus, the purpose of this study was to verify possible differences in immune-mediated inflammatory parameters, evaluated by the expression of toll-like receptors (TLRs), and oxidative stress [evaluated by nuclear factor kβ (NF-kβ)] in a group of newly diagnosed hypertensive patients with or without 1-h post-load hyperglycemia.

## 2. Results

### 2.1. Study Population

Hypertensive patients (*n* = 50) were further divided into two groups on the basis of 1-h post-load plasma glucose: hypertensives with (NGT 1-h ≥ 155, *n* = 25) or without (NGT 1-h < 155, *n* = 25) 1-h post-load hyperglycemia. In Table 1 and Table 2 we reported the baseline characteristics of the study population; comparisons have been made between the control group and the whole group of hypertensive patients, and between the two groups of hypertensive patients. Hypertensives, in comparison with the control group, were older and had a significantly higher body mass index (BMI), systolic blood pressure (SBP), diastolic blood pressure (DBP), fasting glucose, glucose-T60, glucose-T120, fasting insulin, insulin-T60, insulin-T120, homeostasis model assessment (HOMA)-index, uric acid, total cholesterol, low-density lipoprotein (LDL)-cholesterol, triglyceride, and creatinine. In addition, as expected, hypertensive patients had a significantly lower Matsuda index, (estimated glomerular filtration rate (e-GFR), high-density lipoprotein (HDL)-cholesterol, and statistically higher values of TLR2, TLR4, NF-kβ, IL-1β, IL-6, IL-8, IL-10, TNF-α, high-sensitivity C-reactive protein (hs-CRP), and fibrinogen.

When considering hypertensive patients, NGT 1-h > 155, in comparison with NGT 1-h < 155, had significantly higher values of fasting glucose, glucose-T60, glucose-T120, fasting insulin, insulin-T60, insulin-T120, HOMA-index, uric acid, total cholesterol, LDL-cholesterol, triglyceride, TLR2, TLR4, NF-kβ, IL-1β, IL-6, IL-8, TNF-α, and hs-CRP. Obviously, we also documented significantly lower values of the Matsuda index, e-GFR, and HDL-cholesterol. No significant differences were observed in fibrinogen values.

In Figure 1 we graphically reported both glucose and insulin curves during OGTT. The calculation of the area under the curve (AUC) to compare these biological responses resulted in a significant increase in both glucose and insulin AUC in NGT 1-h ≥ 155 in comparison with normal and NGT 1-h < 155. No significant differences were observed comparing the NGT 1-h < 155 and normal groups.

In Figure 2 we graphically reported cytokine values in the four groups; it is evident that hypertensives showed almost doubled values of cytokines when compared to the control group. The same trend was observed when comparing NGT ≥ 155 to NGT < 155, confirming a different inflammatory burden among groups.

### 2.2. Correlational Analysis

At linear correlational analysis (Table 3), in the whole study population, we observed that the major independent covariates related to TLRs and NF-kβ were the Matsuda index, 1-h post-load glycemia, BMI, systolic blood pressure (SBP), age, hypertensive status, and e-GFR. Similar results were observed in NGT 1-h ≥ 155 in comparison with NGT 1-h < 155.

At the subsequent stepwise multivariate linear regression analysis (Table 4), including the variables reaching statistical significance in univariate models, in the whole study population, as well as in both hypertensive groups, we observed that the Matsuda index was retained as the strongest predictor of TLRs and NF-kβ, explaining about 46% of their variation; interestingly, 1-h post-load glycemia was also retained as an independent covariate in all analysis models. In addition, it is clinically relevant to remark that the final models retained other metabolic covariates and e-GFR, all factors associated with low-grade subclinical inflammation.

## 3. Discussion

The novelty of this study is the demonstration that newly diagnosed hypertensive patients, in comparison with a control group, have a worse inflammatory profile and greater activation of the innate immune system. But the most relevant and still not investigated finding of this study is that, when considering only hypertensive patients, NGT 1-h ≥ 155 subjects are also characterized by a higher expression of inflammatory molecules in comparison with NGT 1-h < 155 ones. In fact, they show significantly higher levels of TLRs, NF-kβ, some cytokines, and the hs-CRP, confirming that this inflammatory condition is due to the activation of the innate immune system. All these data show that the stratification of hypertensive patients, based on a specific phenotype with hyperglycemia at 1-h during the OGTT, allows us to identify a different risk profile that goes beyond the increase in BP. Taken together, present data contribute to expanding the pathophysiological role of the immune system and its related inflammatory cascade, whose activation has been considered, for a long time, as the organism’s helpful response to reduce cellular and organ damage against many endogenous and exogenous danger signals [23,24]. On the contrary, present findings confirm that inflammatory immune response is the most important pathogenetic mechanism operating in the appearance and progression of some chronic and non-communicable diseases [9,11,12].

Clinically relevant, our data demonstrate that the major and independent predictor of the activation of the TLRs in the whole study population as well as in both groups of hypertensive patients is the Matsuda index, a reliable marker of insulin resistance [25]; in this context, it is also useful to highlight that other metabolic covariates, as well as e-GFR, were retained in the final models of the multivariate analysis, all conditions associated with some degree of both insulin resistance and low-degree of subclinical inflammation [7,26,27,28,29]. This evidence is not surprising, as it is well established that hypertensive patients have a certain degree of insulin resistance [30] that participates in the appearance and progression of subclinical organ damage [31], an intermediate step in the development of the CV continuum. Notably, our data demonstrate that this condition of insulin resistance is also strongly present in NGT 1-h ≥ 155 subjects who, according to current guidelines, we wrongly continue to consider normotolerant and, therefore, at low risk of both incident diabetes and CV events [32]. According to this, the major clinically and prognostically relevant finding of this study is that 1-h post-load hyperglycemia was retained as an independent predictor of TLRs and NF-kβ, together with the Matsuda index in the same model. On this basis, it would be useful to redefine current criteria for the diagnosis of type 2 diabetes mellitus by also taking into account 1-h post-load glucose values. In fact, these findings contribute to better characterizing this specific phenotype that causes multiple subclinical organ damage at cardiac, vascular, and renal levels [18,19,20,21], and a higher risk of incident diabetes as previously demonstrated [15,16,17,18,19,20,21,22]. Consequently, to optimize the treatment of essential hypertension, it is necessary to: 1. routinely detect any other metabolic alteration present in patients with elevated BP; 2. stratify the global CV risk in each patient by identifying the possible presence of subclinical organ damage that negatively impacts on its prognosis; 3. personalize the therapy in individual patients considering, in addition to the hemodynamic overload, any other coexisting metabolic alterations. In fact, only in this way will it be possible to obtain a real benefit in terms of the reduction of risk of future fatal and non-fatal CV events. These considerations reinforce the concept, affirmed for years but never really implemented, of a global risk stratification in a holistic view of the patient that goes beyond the single organ pathology, a medicine of the person, and not the management of single organ damage.

Another relevant finding highlighted by the present study is the close relationship between glycometabolic alterations and NF-kβ. The latter, in fact, is a transcriptional factor that plays a primary role in the regulation of the immune response and its associated inflammation, and in the proliferative process [33,34]; all these elements are involved in the appearance and progression of vascular atherosclerotic disease, first, and CV events, later. This condition is also facilitated by the increase in oxidative stress induced by the activation of NF-kβ, which reduces the bioavailability of nitric oxide that is able to preserve the physiological function of endothelial cells that, when they become dysfunctional, represent a powerful and independent predictor of new cardiovascular events [35,36]. Thus, according to this, it is possible to affirm that present data expands our knowledge of the pathophysiological mechanisms involved in the determinism of vascular damage induced by CV risk factors. Relevant in this context is the role of both TNF-α and IL-6 that induce an altered expression of insulin receptors with consequent worsening of insulin resistance status and associated hyperglycemia [37,38,39]. Obviously, the persistence of these mechanisms activates a reverberant circuit that contributes to the persistence of the metabolic and hemodynamic alterations. In fact, it is well established that insulin resistance is associated with the overactivation of the sympathetic nervous system, with consequent expansion of the circulating blood volume due to the increased renal hydrosaline reabsorption, and increase in vascular resistance [40].

In conclusion, the results of this study contribute to a better definition of the pathophysiological mechanisms by which the classic CV risk factors operate in the induction and progression of vascular damage. In addition, they strengthen the concept that CV risk factors can, very often, cluster and activate bidirectional mechanisms, with consequent persistence of the same and amplification of the negative effects on the appearance and progression of subclinical organ damage. Finally, in the absence of therapeutic strategies able to interfere with the immune-mediated inflammation mechanisms, it would be desirable to implement solid interventions of primary and secondary prevention to prevent/reduce vascular damage and consequent fatal and non-fatal CV events.

This study has several limitations, the most important of which is the small sample size. Anyway, given the fact that this is a pathophysiological study, we think that the results obtained could be pivotal for further investigations. Another limitation of this study is the fact that we tested only a small number of inflammatory cytokines and oxidative stress molecules; further studies are required to test a wider panel of cytokines and oxidative stress molecules.

## 4. Materials and Methods

### 4.1. Study Population

For this study, from a very large cohort of Caucasian outpatients who have undergone OGTT and are participating in the CAtanzaro MEtabolic RIsk factors (CATAMERI) study [41], we selected a control group comprising 25 normotensive and NGT subjects (12 males and 13 females, mean age 39.4 ± 11.3 years) and 50 newly diagnosed never-treated hypertensive patients (20 males and 30 females, mean age 56.6 ± 7.1 years). All participants underwent physical examination and a review of their medical history. Causes of secondary hypertension were excluded by a standard clinical protocol, including measurement of plasma renin activity, aldosterone, Doppler studies of the renal arteries, and/or renal scintigraphy or renal angiography. Other exclusion criteria were history or clinical evidence of coronary, valvular heart disease, congestive heart failure, hyperlipidemia, hyperuricemia, peripheral vascular disease, chronic gastrointestinal diseases associated with malabsorption, chronic pancreatitis, history of any malignant or autoimmune disease, history of alcohol or drug abuse, liver or kidney failure (serum creatinine > 1.5 mg/dL and without proteinuria on the dipstick test), and treatments able to modify glucose metabolism. At the time of the first evaluation, participants underwent anthropometrical evaluation (weight, height, and BMI), and routine blood tests.

The Ethical Committee approved the protocol (protocol code 2012.63, date of approval 23 October 2012), and informed written consent was obtained from all participants. All the investigations were performed in accordance with the principles of the Declaration of Helsinki.

### 4.2. Blood Pressure Measurements

Clinic blood pressure (BP), after an initial evaluation at both upper arms, was measured in the arm with higher BP values of the sitting patients, after 5 min of quiet rest, with a validated sphygmomanometer. SBP and DBP were recorded at the first appearance (phase I) and the disappearance (phase V) of Korotkoff sounds. A minimum of three BP readings were taken on three separate occasions at least 2 weeks apart. Baseline BP values were the average of the last two of the three consecutive measurements obtained at intervals of 3 min. Patients with a clinic SBP > 140 mmHg and/or DBP > 90 mmHg were defined as hypertensive [42].

### 4.3. Laboratory Determinations

All laboratory measurements were performed after 12 h of fasting. Plasma glucose was determined immediately by the glucose oxidation method [Glucose analyser, Beckman Coulter, Milan; intra-assay coefficient of variation (CV) 2.2%, inter-assay CV 3.8%]. Total, LDL-, and HDL-cholesterol and triglyceride concentrations were measured by enzymatic methods (Roche Diagnostics GmbH, Mannheim, Germany). Uric acid and creatinine were measured using the Jaffe methodology. Values of estimated glomerular filtration rate (e-GFR) (mL/min/1.73 m^2^) were calculated by using the CKD-EPI equation [43]. High-sensitivity C-reactive protein (hs-CRP) levels were measured with an automated instrument (Cardio—Phase hs-CRP, Siemens Healthcare, Milano, Italy). Fibrinogen was dosed with a coagulation method on the instrument BCSXP. The complete blood count was performed by flux cytometry.

### 4.4. Oral Glucose Tolerance Tests

After 12-h fasting, a 75 g OGTT was performed with 0, 30-, 60-, 90-, and 120-min sampling for plasma glucose and insulin measurements. Glucose tolerance status was defined based on OGTT using the World Health Organization (WHO) criteria. Insulin sensitivity was evaluated using the Matsuda index:

[insulin sensitivity index (ISI)], calculated as follows: 10,000/square root of [fasting glucose (mmol per liter)×fasting insulin (μU per liter)] × [mean glucose×mean insulin during OGTT] [25].

The Matsuda index is strongly related to the euglycemic hyperinsulinemic clamp that represents the gold standard test for measuring insulin sensitivity.

### 4.5. Evaluation of TLR-2 and TLR-4 Expression

Peripheral blood mononuclear cells (PBMCs) were isolated from fresh peripheral samples by Ficoll gradient according to the manufacturer’s instructions within 2 h after blood collection. Isolated PBMCs were harvested and suspended in phosphate-buffered saline (PBS) for TLRs expression analysis by FACS, or in hypotonic buffer, and further processed to evaluate NF-kβ (p65) activity. Isolated PBMCs were labeled with an anti-CD14 antibody conjugated with phycoerythrin (PE) to identify the monocytes within the isolated population, and with an anti-TLR2 antibody conjugated with fluorescein isothiocyanate (FICT), or with an anti-TLR4 antibody conjugated with allophicocyanine (APC). An ISO CD-14 (PE) non-specific antibody was employed as a control isotype to exclude non-specific bonds. Samples were acquired with a FACSCalibur (BD Biosciences) and analyzed by Flow JO software (BD Biosciences). The monocyte population was identified by both forward (FSC) and side scatter (SSC). TLRs expression was considered as the ratio of mean fluorescence intensity (MFI) of the sample and MFI of the isotype control.

### 4.6. NF-kβ (p65) Activity Assay

To evaluate NF-kβ (p65) activity, PBMCs were lysed with a hypotonic buffer containing 10 mM Hepes, 0.1 mM EDTA, and phosphatase inhibitors. Supernatants containing cytosolic fractions were removed, and pellets were resuspended in a nuclear extraction buffer containing 10 mM Hepes, 0.1 mM EDTA, 1.5 mM MgCl_2_, 420 mM NaCl, and 10% glycerol. NF-kβ activity in nuclear lysates was then measured with NF-kβ specific ELISA assay (Cayman Chemical Company, Ann Arbor, MI, USA), according to the manufacturer instructions. A positive control was included in each ELISA plate; samples were assessed in duplicate.

### 4.7. Pro-Inflammatory Cytokines Measurement

To assess cytokines levels, serum samples were loaded into a biochip array containing specific primary antibodies (Cytokine and Growth Factors Array, Randox Laboratories, Crumlin, UK) and processed according to the user manual. Briefly, biochips were incubated for 1 h at 37 °C, washed and incubated for an additional hour with the HRP-conjugated secondary antibody. Biochips were acquired with a biochip reader and analyzed with dedicated software. The assay measuring range was 0.0–2.4 pg/mL for IL-1β; 0.0–5.6 pg/mL for IL-6; 1.9–17.4 pg/mL for IL-8; 0.0–6.3 pg/mL for IL-10; 0.0–13.3 pg/mL for TNF-α. Intra and inter assay precision values were respectively 8.1% and 9.4% for IL-1β, 7.3% and 12.9% for IL-6; 7.7% and 6.7% for IL-8; 7.8% and 5.2% for IL-10, and 9.9% for TNF-α.

### 4.8. Statistical Analysis

Descriptive data of normally distributed variables are reported as the mean ± SD, while binary data as percent frequency. Distribution normality was assessed with the Shapiro–Wilk test. Differences between groups were compared using an unpaired *t*-test when clinical and biological data were expressed as continuous variables, and the χ^2^ test for categorical variables. Relationships between paired parameters were analyzed by the Pearson product moment correlation coefficient. Linear regression analysis was performed to test the relationship between TLRs and NF-kβ with some independent covariates (age, gender, BMI, SBP, LDL- and HDL-cholesterol, triglyceride, insulin sensitivity indices, 1-h post-load glycemia, creatinine, e-GFR, and hypertensive status). Successively, variables reaching statistical significance and gender, as dichotomic value, were inserted in a stepwise multivariate linear regression model to determine the independent predictors of TLRs and NF-kβ in the whole study population and in the two groups of NGT hypertensive patients. Differences were assumed to be significant at two-tailed *p* values < 0.05. All calculations were done with a standard statistical package (SPSS for Mac version 21.0, Chicago, IL, USA).

## Figures and Tables

**Figure 1 ijms-23-10891-f001:**
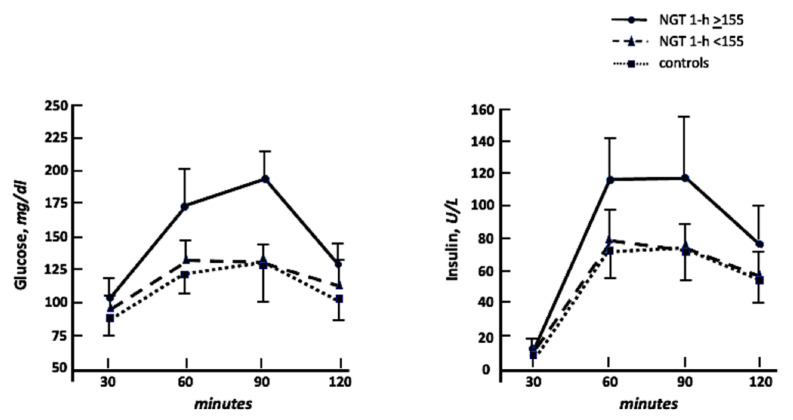
Glucose and insulin curves during oral glucose tolerance test in NGT 1-h ≥ 155, NGT 1-h < 155, and controls.

**Figure 2 ijms-23-10891-f002:**
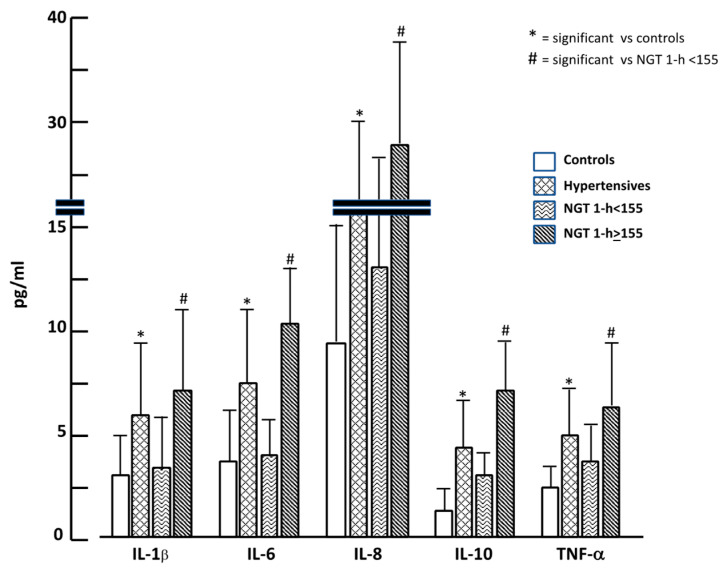
Inflammatory cytokine values in the four study groups.

**Table 1 ijms-23-10891-t001:** Anthropometric, biochemical, and hemodynamic characteristics of the whole study population and of the two groups of hypertensive patients stratified by 1-h post-load plasma glucose.

Variables	Controls (*n* = 25)	HT (*n* = 50)	*p*	NGT 1-h < 155 (*n* = 25)	NGT 1-h ≥ 155 (*n* = 25)	*p*
Gender, M/F	12/13	20/30	0.324	16/9	14/11	0.564
age, years	39.4 ± 11.3	56.6 ± 7.1	0.000	56.0 ± 7.3	57.3 ± 7.1	0.506
BMI, Kg/m^2^	23.9 ± 2.4	27.6 ± 2.1	0.000	27.3 ± 1.5	27.9 ± 2.6	0.323
SBP, mmHg	117.0 ± 8.7	146.0 ± 5.4	0.000	146.0 ± 4.5	146.0 ± 6.3	0.979
DBP, mmHg	71.7 ± 6.7	91.8 ± 8.6	0.000	91.0 ± 11.5	92.5 ± 11.5	0.517
Glucose, mg/dL	86.6 ± 11.4	98.8 ± 10.5	0.000	95.6 ± 8.6	102.0 ± 11.4	0.029
Glucose T_60_, mg/dL	130.4 ± 31.4	162.0 ± 36.7	0.000	130.0 ± 7.6	193.9 ± 23.5	0.000
Glucose T_120_, mg/dL	102.0 ± 17.9	121.0 ± 16.9	0.000	114.0 ± 15.6	128.0 ± 15.4	0.003
Insulin, U/L	9.9 ± 2.1	10.9 ± 3.9	0.241	10.2 ± 2.5	11.7 ± 4.9	0.187
Insulin T_60_, U/L	74.9 ± 19.9	96.3 ± 39.8	0.014	74.4 ± 14.9	118.2 ± 44.9	0.000
Insulin T_120_, U/L	54.5 ± 14.8	67.5 ± 26.0	0.023	57.4 ± 15.4	77.6 ± 30.5	0.005
HOMA	2.1 ± 0.6	2.7 ± 1.0	0.014	2.4 ± 0.7	2.9 ± 1.1	0.060
Matsuda index	86.9 ± 19.6	65.7 ± 20.0	0.000	76.7 ± 15.5	54.7 ± 18	0.000
Creatinine, mg/dL	0.77 ± 0.09	0.84 ± 0.09	0.006	0.84 ± 0.09	0.85 ± 0.10	0.518
e-GFR, mL/min/1.73 m^2^	112.1 ± 17.7	96.3 ± 16.9	0.003	106.4 ± 16.6	91.5 ± 14.8	0.002
Uric acid, mg/dL	4.3 ± 0.9	5.1 + 1.0	0.001	4.6 ± 0.5	5.7 ± 1.0	0.000
Cholesterol, mg/dL	190.5 ± 34.8	212.0 ± 14.0	0.003	188.1 ± 28.5	220.2 ± 17.8	0.000
LDL-Chol, mg/dL	107.5 ± 34.1	138.4 ± 17.2	0.000	105.1 ± 26.5	146.0 ± 17.1	0.001
HDL-Chol, mg/dL	63.6 + 9.5	49.2 ± 9.5	0.000	46.6 ± 8.5	47.8 ± 6.6	0.245
Triglyceride, mg/dL	99.8 ± 40.9	122.3 ± 34.7	0.015	112.4 ± 30.4	132.3 ± 36.2	0.041

HT: hypertensives; NGT: normal glucose tolerance; BMI: body mass index; SBP: systolic blood pressure; DBP: diastolic blood pressure; HOMA: homeostasis model assessment; e-GFR: estimated-glomerular filtration rate; LDL: low density lipoprotein; HDL: high density lipoprotein.

**Table 2 ijms-23-10891-t002:** Inflammatory parameters of the whole study population and of the two groups of hypertensive patients stratified by 1-h post-load plasma glucose.

Variables	Controls (*n* = 25)	HT (*n* = 50)	*p*	NGT 1-h < 155 (*n* = 25)	NGT 1-h ≥ 155 (*n* = 25)	*p*
TLR2, MFI	4.6 ± 2.4	7.7 ± 3.9	0.000	5.9 ± 2.6	9.4 ± 4.2	0.002
TLR4, MFI	7.1 ± 4.3	10.4 ± 4.1	0.002	7.8 ± 2.3	13.1 ± 3.9	0.000
NF-kβ	0.07 ± 0.05	0.18 ± 0.06	0.000	0.14 ± 0.04	0.21 ± 0.07	0.000
IL-1β, pg/mL	2.8 ± 1.7	5.9 ± 2.8	0.000	3.2 ± 2.1	6.9 + 3.4	0.000
IL-6, pg/mL	3.1 ± 2.3	7.5 ± 3.9	0.000	4.1 ± 1.6	10.8 + 2.6	0.000
IL-8, pg/mL	9.1 ± 6.0	19.7 ± 10.3	0.000	13.3 ± 5.6	27.6 + 9.3	0.000
IL-10, pg/mL	1.6 ± 0.9	4.5 ± 2.4	0.000	2.8 ± 1.3	6.8 + 2.5	0.000
TNF-α, pg/mL	2.5 ± 1.2	5.0 ± 2.6	0.000	3.3 ± 1.4	6.4 ± 2.9	0.000
hs-CRP, mg/dL	1.4 ± 0.9	3.7 ± 1.6	0.000	2.7 ± 1.0	4.8 ± 1.5	0.000
Fibrinogen, mg/dL	263.6 ± 35.8	301.4 ± 49.2	0.001	301.4 ± 38.3	301.6 ± 59.1	0.111

HT: hypertensives; NGT: normal glucose tolerance; TLR2: toll-like receptor-2; TLR4: toll-like receptor-4; NF-kβ: nuclear factor-kb; IL-1β: interleukin-1b; IL-6: interleukin-6; IL-8: interleukin-8; IL-10: interleukin-10; TNF-α: Tumor necrosis factor-α; hs-CRP: high-sensitivity C-reactive protein; MFI: mean fluorescence intensity.

**Table 3 ijms-23-10891-t003:** Independent covariates related to TLRs and NF-kβ expression in the whole study population, in hypertensives NGT 1-h < 155, and in NGT 1-h ≥ 155.

Variables	TLR2		TLR4		NF-kβ	
	R	P	R	P	R	P
** *Whole study population* **						
Matsuda index	−0.686	0.0001	−0.684	0.0001	−0.676	0.0001
1-h post-load glycemia	0.583	0.0001	0.615	0.0001	0.632	0.0001
BMI	0.501	0.0001	0.430	0.0001	0.552	0.0001
SBP	0.480	0.0001	0.427	0.0001	0.643	0.0001
Age	0.453	0.0001	0.456	0.0001	0.558	0.0001
HT vs. NT status	0.413	0.0001	0.356	0.001	0.664	0.0001
e-GFR	−0.425	0.0001	−0.416	0.0001	−0.570	0.0001
LDL-cholesterol	0.414	0.0001	0.543	0.0001	0.547	0.0001
Triglyceride	0.284	0.007	0.359	0.001	0.245	0.017
HDL-cholesterol	−0.260	0.012	−0.287	0.0006	−0.474	0.0001
Gender, M vs. F	−0.155	0.092	−0.079	0.250	−0.074	0.265
Creatinine	0.078	0.254	0.048	0.340	0.318	0.003
** *Hypertensives NGT-1h < 155* **						
Matsuda index	−0.593	0.001	−0.528	0.003	−0.610	0.001
LDL-cholesterol	0.363	0.037	0.062	0.385	0.302	0.071
Age	0.319	0.060	0.009	0.483	0.322	0.058
HDL-cholesterol	−0.232	0.132	−0.035	0.433	−0.338	0.049
SBP	0.209	0.158	0.325	0.056	0.502	0.005
Triglyceride	0.202	0.166	0.487	0.007	0.069	0.372
Creatinine	0.185	0.187	0.331	0.053	0.081	0.351
1-h post-load glycemia	0.170	0.208	0.517	0.004	0.214	0.152
Gender, M vs. F	−0.163	0.219	0.105	0.309	0.176	0.200
BMI	0.102	0.313	−0.207	0.161	0.271	0.095
e-GFR	−0.008	0.485	−0.257	0.108	−0.063	0.382
** *Hypertensives NGT 1-h ≥ 155* **						
Matsuda index	−0.709	0.0001	−0.630	0.0001	−0.327	0.055
LDL-cholesterol	0.142	0.250	0.055	0.397	0.124	0.278
Age	0.076	0.360	0.106	0.308	−0.009	0.483
HDL-cholesterol	−0.028	0.447	0.001	0.497	0.068	0.374
SBP	0.345	0.045	0.283	0.085	0.052	0.403
Triglyceride	0.200	0.169	0.072	0.366	0.276	0.091
Creatinine	0.339	0.048	0.368	0.035	0.229	0.136
1-h post-load glycemia	0.441	0.014	0.381	0.030	0.118	0.287
Gender, M vs. F	0.532	0.003	−0.417	0.019	−0.391	0.027
BMI	0.498	0.006	0.322	0.058	0.132	0.265
e-GFR	−0.236	0.128	−0.157	0.227	−0.524	0.004

BMI: body mass index; SBP: systolic blood pressure; e-GFR: estimated-glomerular filtration rate; LDL: low-density lipoprotein; HDL: high-density lipoprotein; HT vs. NT status: hypertensive vs. normotensive status.

**Table 4 ijms-23-10891-t004:** Multiple regression analysis in the whole study population, in hypertensives NGT 1-h < 155, and NGT 1-h ≥ 155.

** *Whole Study Population* **			
**TLR2**	**r^2^ Partial**	**r^2^ Total**	**P**
Matsuda index	47.0%	47.0%	0.0001
1-h post-load glycemia	4.6%	51.6%	0.015
Age	3.6%	55.2%	0.020
**TLR4**			
Matsuda index	46.8%	46.8%	0.0001
LDL-cholesterol	7.9%	53.4%	0.002
1-h post-load glycemia	4.2%	57.6%	0.006
**NF-kβ**			
Matsuda index	45.7%	45.7%	0.0001
HT vs. NT status	16.1%	61.8%	0.0001
e-GFR	8.0%	69.8%	0.0001
Triglyceride	3.3%	73.1%	0.003
1-h post-load glycemia	2.1%	75.2%	0.020
** *Hypertensives NGT-1h < 155* **			
**TLR2**			
Matsuda index	35.2%	35.2%	0.002
**TLR4**			
Matsuda index	27.9%	27.9%	0.007
1-h post-load glycemia	12.5%	40.4%	0.042
Triglyceride	11.0%	51.5%	0.041
**NF-kβ**			
Matsuda index	37.2%	37.2%	0.001
** *Hypertensives NGT-1h ≥ 155* **			
**TLR2**			
Matsuda index	50.2%	50.2%	0.0001
1-h post-load glycemia	27.6%	77.8%	0.0001
BMI	5.0%	82.8%	0.0001
**TLR4**			
Matsuda index	39.7%	39.7%	0.0001
1-h glycemia	28.5%	682%	0.0001
Triglyceride	9.3%	77.5%	0.008
**NF-kβ**			
e-GFR	27.5%	27.5%	0.007

BMI: body mass index; e-GFR: estimated-glomerular filtration rate; LDL: low-density lipoprotein; HT vs. NT status: hypertensive vs. normotensive status.

## Data Availability

The datasets generated during and analyzed in the current study are available from the corresponding author upon reasonable request.

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
