# Peer review of "Immuno-Mediated Inflammation in Hypertensive Patients with 1-h Post-Load Hyperglycemia"

_ijms, 2022, doi:10.3390/ijms231810891_

Round 1

Reviewer 1 Report

This original paper present an interesting study, which have important clinical implications.  The authors have tested the differences in immune-mediated inflammatory parameters in newly diagnosed hypertensives with or without 1-hour post-load hyperglycemia. It is well-known that inflammatory markers may play a major adjunctive role in the assessment of cardiovascular risk in selected patients, and may be considered as emergent therapeutic targets. The manuscript is well written and well designed.

General comments

I want to underline that, at list 13 references (out of 37) - which means one third of references, were signed by the same group of authors. The references 12-17 signed by the same group of authors were published between 2010-2015; three out of these (references 13,14 and 15) are about one hour post-load plasma glucose levels in essential hypertension. In addition, institutional review board statement was approved by the Ethics Committee in 2012. This group of authors are intensively preoccupied by this theme.

Minor revisions

How do you choose the inflammatory cytokines and oxidative stress molecules that were used in this study?

In table 3, in hypertensives NGT 1-h>155 group, gender, M vs F was negative significantly correlated with the covariates TLRs and NF-kβ. On the contrary, in hypertensives NGT-1h<155 group was not significantly correlated with these covariates. How do you explain this? What means these gender differences?

 I suggest putting the variables for this table in the same order for hypertensives NGT 1-h>155 and hypertensives NGT-1h<155 group. It is difficult to compare these variables.

Page 2 lines 46-50: The authors cited only their papers. Please cite other paper as you have said: ``In the last years we and others contributed to define a specific phenotype.``

Page 9 line 270: The number of patients in the three groups of the study was the same (n=25). How do you selected these patients?

Page 10 line 289: Blood pressure was measured in the left arm. Usually for newly diagnosed hypertensive patients blood pressure must be assessed on both arms. I wondering if it is appropriate this manner of blood pressure assessment. Why the left and not the right arm? What do you think about this?

Reviewer 2 Report

he study Immuno-mediated inflammation in hypertensive patients with 21-hour post-load hyperglycemia” by Pertisone et al. investigated the possible differences in immune-mediated inflammatory parameters innewly-diagnosed hypertensives with or without hour post-load hyperglycemia. 

I have several comments regarding this study:  

Major 

Statistical analysis has to be re-done. Comparisons must be made between control and the two hypertensive groups: NGT 1-h>155 and NGT<155 groups. 

There are the same typos errors throughout the manuscript. 

Reviewer 3 Report

Reviewer comments and suggestions

The authors in this study reported the possible differences in immune-mediated inflammatory parameters in newly diagnosed hypertensives with or without 1-hour post-load hyperglycemia. For this study, they recruited 25 normotensives (NGT) and 50 hypertensives normo-tolerant at oral glucose tolerance test, categorised into two groups based on 1-hour post-load plasma glucose: NGT 1-h>155 (n= 25) and NGT 1-h<155 (n= 25)><155 (N=25). The result of the study reported was higher BMI, creatinine and inflammatory parameters in hypertensives in contrast to controls.

Moreover, NGT 1-h>155 had a worse glycometabolic profile and higher values of TLR2, TLR4, NF-kb, IL-1b, IL-6, IL-8, TNF-a and hs- CRP in comparison with NGT 1-h<155. Matsuda-index and 1-h post-load 27 glycemia were retained as major predictors of TLRs and NF-kb. These results contribute to better 28 characterize cardiovascular risk in hypertensives.><155.

Overall, the manuscript needs a thorough revision for publication. Few concerns are below to be incorporated in the revised version of the manuscript. 

  1. Line 36-38, These lines required suitable reference.
  2. Line 42 reference 5 Please explore the lines
  3. Line 45 The authors put so many references at a place, that it is not professionally correct. Better to check and describe a few of them to present your solid point in the manuscript that why this study was needed
  4. How this glucose category was defined, please explain this as well
  5. Line 48-49 These studies need to be explored
  6. Study results section, 2.1 I think if the authors pointout the values in the table so no need to add the same values for increasing the unnecessary text.
  7. Line 103, This result should be well defined with the help of figures, you can make 30, 60, 120 min interval data.
  8. For table 3, The representation was not good, you can make a correlation table for this if possible. It’s my suggestion
  9. Discussion It would be nice to add up the novelty of the study here, in the first para of the discussion
  10. Line 205-207 How the authors put references here, please check it
  11. Line 219 is not a good word of choice “and a higher risk of incident diabetes [17] as previously demonstrated by us.”
  12. Line 237-238 The next point of endothelial dysfunction was missing, please explain
  13. Line 284-285 please add the Ethical approval number

Round 2

Reviewer 2 Report

The Authors have satisfactorily addressed all of my concerns.